# Consensus, Polarization and Hysteresis in the Three-State Noisy *q*-Voter Model with Bounded Confidence

**DOI:** 10.3390/e24070983

**Published:** 2022-07-16

**Authors:** Maciej Doniec, Arkadiusz Lipiecki, Katarzyna Sznajd-Weron

**Affiliations:** Department of Theoretical Physics, Wrocław University of Science and Technology, 50-370 Wrocław, Poland; 250206@student.pwr.edu.pl (M.D.); lipiecki.arkadiusz@gmail.com (A.L.)

**Keywords:** agent-based model, opinion dynamics, bounded confidence, voter model, consensus, polarization, hysteresis, tipping point, bifurcation

## Abstract

In this work, we address the question of the role of the influence of group size on the emergence of various collective social phenomena, such as consensus, polarization and social hysteresis. To answer this question, we study the three-state noisy *q*-voter model with bounded confidence, in which agents can be in one of three states: two extremes (leftist and rightist) and centrist. We study the model on a complete graph within the mean-field approach and show that, depending on the size *q* of the influence group, saddle-node bifurcation cascades of different length appear and different collective phenomena are possible. In particular, for all values of q>1, social hysteresis is observed. Furthermore, for small values of q∈(1,4), disagreement, polarization and domination of centrists (a consensus understood as the general agreement, not unanimity) can be achieved but not the domination of extremists. The latter is possible only for larger groups of influence. Finally, by comparing our model to others, we discuss how a small change in the rules at the microscopic level can dramatically change the macroscopic behavior of the model.

## 1. Introduction

Over the past decade, there has been a significant increase in the number of articles devoted to the dynamics of opinion. According to the Scopus data base, the total number of papers in which the phrase “opinion dynamics” appears in the title, abstract or keywords is approaching 2000, of which as many as 250 were published last year! Among all these 2000 articles, 24.7% have been published in the area of computer science, 24.1% in mathematics, 13.5% in physics, 5.1% in social science, 3.4% in decision science and the rest in other subject areas. Of course, not all papers on opinion dynamics include the phrase “opinion dynamics” in the title/summary/keywords; therefore, the numbers we provided are certainly undercounted.

However, they show that this is an area that is attracting increasing attention from researchers in a variety of disciplines. The popularity of the topic is also demonstrated by the number of recent review articles on opinion dynamics [1,2,3,4,5,6,7]. Most opinion models can be classified into one of two main families: continuous or discrete opinion models [2,3,4,5]. The second family is dominated by models with binary opinions; however, some multi-state models have also been proposed [8,9,10,11,12,13,14,15,16,17,18,19,20,21,22,23,24]. Those with three opinions [9,10,12,14,16,17,21,22,25,26,27,28,29,30,31,32], often interpreted as leftists, centrists and rightists, are particularly relevant in the context of this work.

The one we propose and study here is the modification of the recently introduced multi-state noisy *q*-voter model (qVM) [24]. The modification involves the introduction of a bounded confidence rule (BC), which was originally introduced within continuous opinion models [1,33,34,35].

In the framework of discrete models, this rule typically involves prohibiting interactions between agents whose opinions differ by more than one [8,9,10,14,36]. This means, for example, that leftists and rightists do not interact [10]. Recently, the BC rule has also been extended to other types of social response, such as independence, which means that no change in the state of an agent can be greater than one during a single update [36]. We will not use this extended definition of BC here for reasons that will be discussed in the last section of this paper. This means that, within our model, BC applies exclusively to the interactions between agents and independence plays the role of pure noise, as in the noisy multi-state qVM without BC [24].

One might have doubts: if there are so many different models, why introduce yet another one? The answer lies in a characteristic feature of the *q*-voter model, which allows determination of the role of the size of the influence group on the evolution and stationary states of the system. Here, another question arises: why is it important to examine the role of group size? This is because a number of empirical studies showed that the group size makes a difference in many aspects [37,38,39,40,41].

For example, it was shown that conformity increases with the size of the group of influence, however, only to the certain threshold of four to five people [37]. Moreover, it was shown that the average conversation group size is around three [40], and groups containing three to six members were significantly more productive and creative than larger groups [38]. Here, we ask the question whether the size of the influence group can influence the emergence of polarization, consensus or another type of behavior. To our knowledge, such a question has not yet been asked.

The special feature of the qVM, mentioned above, is that the size of the influence group *q* is a parameter of the model and is not equivalent to the total number of friends (neighbors) of a given agent [42]. The logic behind qVM is the following: even if an agent has many friends, at a given moment, it interacts only with a few of them. We show that a variety of collective social phenomena can be observed depending on the size of the influence group *q*, which could not be observed within the three-state qVM without BC [24] or within the three-state qVM with an extended definition of BC [36].

In particular, we show that, for small values of *q*, polarization, as well as consensus on moderate opinion or disagreement, can be achieved; however, extremism never wins. On the other hand, for larger groups of influence and low independence, consensus on extreme opinion can be achieved. In contrast, in [24,36], polarization cannot be observed for any value of the model parameters. In this work, we use the term consensus as defined by Webster’s or Oxford dictionary, which define it as a general agreement and distinguish it from unanimity.

In fact, consensus, understood as unanimity, is almost impossible to achieve in real-life large social groups. It only appears in theoretical models in the absence of noise. In the presence of noise, only strong dominance of one opinion is possible [24,43], which basically agrees with the actual definition of consensus. Therefore, in the rest of the paper, we use the words consensus and dominance of one opinion interchangeably.

Summarizing, the purpose of this work is twofold:Socially motivated: to determine the role of the size of the influence group on the emerging social behavior (consensus, polarization, hysteresis, etc.).Theoretically motivated: to evaluate the impact of a small change introduced to the model at the microscopic level on its performance at the macroscopic level, since the results for this model without BC [24] and with an extended definition of BC [36] are already known.

The remainder of this paper is structured as follows. In Section 2, we introduce the model and describe an algorithm for an elementary update. In Section 3, we derive, within the mean-field approach (MFA), the set of differential equations that describe the temporal evolution of the system. Moreover, using the symmetry between extreme states, we analytically calculate the stationary behavior of the system. We present the results in Section 4, which is divided into three subsections. In Section 4.1, we present sample trajectories obtained numerically within the MFA equations.

Then, in Section 4.2, we analyze the model more systematically and present the influence of the model parameters on the stationary states. Finally, in Section 4.3 we present phase portraits that help to identify the role of initial conditions and to understand more deeply the rich behavior displayed by the model. We conclude the paper with a discussion of the obtained results in the context of other similar models. This allows us to show how a small change in rules at the microscopic level can dramatically change the macroscopic behavior of the model.

## 2. Model

We consider a society of *N* agents placed at the nodes of an arbitrary, undirected, social network of size *N*. This means that all nodes are occupied and are equivalent to agents. Therefore, we use here the terms node and agent interchangeably. Each node i∈{1,…,N} has a set of $k_{i}$ neighbors—that is, the set of nodes directly linked to node *i*. We assume that the agents do not belong to their own neighborhood, following other versions of qVM [42,44]. Furthermore, as in most previous papers on qVM, we use a random sequential update scheme. This means that, in an elementary time step Δt, we update the state of a single agent (target), which is chosen at random. A time unit that corresponds to the single Monte Carlo step consists, as usual, of *N* elementary updates—that is, NΔt=1. This update scheme mimics continuous time *t* for large systems, since Δt=1/N→0 for N→∞.

Unlike most versions of the qVM, which, as reviewed in [2], describe binary opinions, here an agent can be in one of three alternative states $S_{i}\left(t\right)\in \left\{1,2,3\right\}$. To our knowledge, such non-binary versions of the qVM have been considered so far only in [24,36]. As in many other articles on qVM [3,43,44,45,46,47,48,49,50,51], the state of an agent changes over time under the influence of one of two social responses with complementary probabilities: independence with probability *p* or conformity with probability 1−p. This type of qVM is known as a noisy qVM [24,46,47,50] or a qVM with independence [43,44,45,48,49]. The novelty with respect to the multi-state model proposed in [24] is the introduction of bounded confidence (BC). Previously, the unanimity of the influence group of *q* agents was sufficient to induce conformity. Here, we additionally require that the opinion of the unanimous influence group differs by no more than one level from the opinion $S_{i}\left(t\right)$ of a target, which means that agents with opinions 1 and 3 do not interact. As a result, a single update at time *t* is defined as follows:Choose a target agent i∼U{1,N}, where U{1,N} is a discrete uniform distribution in the interval [1,N],Choose r∼U(0,1), where U(0,1) is a continuous uniform distribution in the interval (0,1), to determine the type of social response,If r<p then *independence*: $S_{i}\left(t+\Delta t\right):=1,2$ or 3 with equal probabilities 1/3,Otherwise *conformity*:
(a)Select at random without repetition *q* agents from $k_{i}$ neighbors of the target agent—they form the source of influence, called also the *q*-panel, and are indexed by j=1,…,q.(b)If the *q*-panel is unanimous, i.e., $\forall_{j=1,\ldots ,q}S_{j}\left(t\right)=S\in \left\{1,2,3\right\}$ and the BC requirement is fulfilled, then $S_{i}\left(t+\Delta t\right)=S$.

## 3. Mean-Field Approach

The model described above can be considered on the top of any undirected graph; however, here, we focus exclusively on the complete graph (CG) for two reasons:For the CG we are able to obtain exact analytical results within MFA.To understand the role of BC in three-state qVM, we need to use the same structure as in [24], in which three-state qVM without BC was considered—that is, CG.

We realize that the structure of the complete graph is realistic only in some real-life cases, such as relatively small social groups. However, we would like to clarify what we mean when we write relatively small, in the context of the qVM. Note that in the qVM, only *q* neighbors influence the agent at one time, no matter how large the neighborhood is. To visualize what this means, let us take the following example. In Warsaw, the capital of Poland, which is our home country, there are over $2\times 10^{5}$ students.

A conversation between any *q* of Warsaw students along the Vistula River, in one of the libraries, at the student party, etc. is possible even if they have never met before. Someone might have the objection that $10^{5}$ is not yet infinity, which is used within MFA. This is definitely true; however, it has been shown in several papers that the results on the complete graphs of size $N=10^{5}$ agree perfectly with the MFA results [24,36,43]. Nevertheless, in the future we also plan to check what is the role of the network structure in such a model and then other structures will be also used.

As usual, we start by writing down the transition probabilities: the probabilities that the state of the system changes from one state to another during the elementary update of duration Δt. For the CG, the state of the system is fully described by the concentration of agents with opinion *k* [2,44]:(1)$$c_{k}=\frac{N_{k}}{N},$$
where $N_{k}$ is the number of agents in the state k∈{1,2,3}. As we use random sequential updates, $c_{k}$ can change only by ±1/N in a single update of duration Δt=1/N. We use the following notation:(2)$$\begin{array}{ccc} \gamma_{k}^{+} & = & Pr\left(c_{k}\left(t+\Delta t\right)=c_{k}\left(t\right)+\frac{1}{N}\right),\\ \gamma_{k}^{-} & = & Pr\left(c_{k}\left(t+\Delta t\right)=c_{k}\left(t\right)-\frac{1}{N}\right). \end{array}$$
We can now write down the time evolution of the expected value $c_{k}$. For N→∞, we can safely assume that the random variable $c_{k}$ converges to its expected value, and thus [2,44]:(3)$$c_{k}\left(t+\Delta t\right)=c_{k}\left(t\right)+\frac{\gamma_{k}^{+}-\gamma_{k}^{-}}{N}.$$
As we assumed that Δt=1/N and N→∞, and we obtain:(4)$$\frac{dc_{k}\left(t\right)}{dt}=\gamma_{k}^{+}-\gamma_{k}^{-}.$$
The state of a system can change due to one of two processes: independence or conformity. Therefore, let us decompose $\gamma_{k}^{\pm }$ into components related to these processes:(5)$$\begin{array}{ccc} \gamma_{k}^{\pm } & = & \gamma_{k,ind}^{\pm }+\gamma_{k,con}^{\pm }. \end{array}$$
The first component, which is related to independence, is straightforward and has the same form for all states *k*—that is,
(6)$$\begin{array}{ccc} \gamma_{k,ind}^{+} & = & \frac{p}{3}\frac{N_{k-1}+N_{k+1}}{N},\\ \gamma_{k,ind}^{-} & = & \frac{2p}{3}\frac{N_{k}}{N}. \end{array}$$
To derive $\gamma_{k,con}^{\pm }$, which is related to conformity, we denote by $k^{\prime }$ the neighboring opinion to the opinion *k*—that is, $k^{\prime }=2$ for k∈{1,3} and $k^{\prime }\in \left\{1,3\right\}$ for k=2. The agent acts as a conformist with probability (1−p) and then changes its state if the group of *q* neighbors shares the same opinion $k^{\prime }$:(7)$$\begin{array}{ccc} \gamma_{k,con}^{+} & = & \left(1-p\right)\sum_{k^{\prime }}\frac{N_{k^{\prime }}}{N}\left(\prod_{i=0}^{q-1}\frac{N_{k}-i}{N-1-i}\right),\\ \gamma_{k,con}^{-} & = & \left(1-p\right)\sum_{k^{\prime }}\frac{N_{k}}{N}\left(\prod_{i=0}^{q-1}\frac{N_{k^{\prime }}-i}{N-1-i}\right). \end{array}$$
Therefore, the explicit forms of $\gamma_{k}^{+}$ can be written as:(8)$$\begin{array}{ccc} \gamma_{1}^{+} & = & \left(1-p\right)\frac{N_{2}}{N}\left(\prod_{i=0}^{q-1}\frac{N_{1}-i}{N-1-i}\right)+\frac{p}{3}\frac{N_{2}+N_{3}}{N},\\ \gamma_{2}^{+} & = & \left(1-p\right)\frac{N_{1}+N_{3}}{N}\left(\prod_{i=0}^{q-1}\frac{N_{2}-i}{N-1-i}\right)+\frac{p}{3}\frac{N_{1}+N_{3}}{N},\\ \gamma_{3}^{+} & = & \left(1-p\right)\frac{N_{2}}{N}\left(\prod_{i=0}^{q-1}\frac{N_{3}-i}{N-1-i}\right)+\frac{p}{3}\frac{N_{1}+N_{2}}{N}, \end{array}$$
Similarly, explicit forms of $\gamma_{k}^{-}$ can be written as follows:(9)$$\begin{array}{ccc} \gamma_{1}^{-} & = & \left(1-p\right)\frac{N_{1}}{N}\left(\prod_{i=0}^{q-1}\frac{N_{2}-i}{N-1-i}\right)+\frac{2p}{3}\frac{N_{1}}{N},\\ \gamma_{2}^{-} & = & \left(1-p\right)\frac{N_{2}}{N}\left(\prod_{i=0}^{q-1}\frac{N_{1}-i}{N-1-i}+\prod_{i=0}^{q-1}\frac{N_{3}-i}{N-1-i}\right)+\frac{2p}{3}\frac{N_{2}}{N},\\ \gamma_{3}^{-} & = & \left(1-p\right)\frac{N_{3}}{N}\left(\prod_{i=0}^{q-1}\frac{N_{2}-i}{N-1-i}\right)+\frac{2p}{3}\frac{N_{3}}{N}. \end{array}$$
In the limit of an infinite system, Equation (Equation 8) can be rewritten as:(10)$$\begin{array}{ccc} \gamma_{1}^{+} & = & \left(1-p\right)\left(c_{2}\cdot c_{1}^{q}\right)+\frac{p}{3}\left(c_{2}+c_{3}\right),\\ \gamma_{2}^{+} & = & \left(1-p\right)\left(c_{1}+c_{3}\right)\cdot c_{2}^{q}+\frac{p}{3}\left(c_{1}+c_{3}\right),\\ \gamma_{3}^{+} & = & \left(1-p\right)\left(c_{2}\cdot c_{3}^{q}\right)+\frac{p}{3}\left(c_{1}+c_{2}\right), \end{array}$$
and Equation (Equation 9) as
(11)$$\begin{array}{ccc} \gamma_{1}^{-} & = & \left(1-p\right)c_{1}\cdot c_{2}^{q}+\frac{2p}{3}c_{1},\\ \gamma_{2}^{-} & = & \left(1-p\right)c_{2}\cdot \left(c_{1}^{q}+c_{3}^{q}\right)+\frac{2p}{3}c_{2},\\ \gamma_{3}^{-} & = & \left(1-p\right)c_{3}\cdot c_{2}^{q}+\frac{2p}{3}c_{3}. \end{array}$$
If we insert transition probabilities (Equation 10) and (Equation 11) to Equation (Equation 4), we obtain the set of equations describing the temporal evolution of the system:(12)$$\begin{array}{ccc} \frac{dc_{1}}{dt} & = & \left(1-p\right)\cdot \left(c_{2}\cdot c_{1}^{q}-c_{1}\cdot c_{2}^{q}\right)+p\left(\frac{1}{3}-c_{1}\right),\\ \frac{dc_{2}}{dt} & = & \left(1-p\right)\cdot \left(\left(c_{1}+c_{3}\right)\cdot c_{2}^{q}-c_{2}\cdot \left(c_{1}^{q}+c_{3}^{q}\right)\right)+p\left(\frac{1}{3}-c_{2}\right),\\ \frac{dc_{3}}{dt} & = & \left(1-p\right)\cdot \left(c_{2}\cdot c_{3}^{q}-c_{3}\cdot c_{2}^{q}\right)+p\left(\frac{1}{3}-c_{3}\right). \end{array}$$
One has to remember that the above equations are not independent, because we deal with the closed system, and thus:(13)$$c_{1}+c_{2}+c_{3}=1.$$

Taking Equation (Equation 13) into account allows us to describe the system using only two independent variables, e.g., $c_{1}$ and $c_{3}$, which reduces Equation (Equation 12) to:(14)$$\begin{array}{ccc} \frac{dc_{1}}{dt} & = & \left(1-p\right)\cdot \left(c_{2}\cdot c_{1}^{q}-c_{1}\cdot c_{2}^{q}\right)+p\left(\frac{1}{3}-c_{1}\right),\\ \frac{dc_{3}}{dt} & = & \left(1-p\right)\cdot \left(c_{2}\cdot c_{3}^{q}-c_{3}\cdot c_{2}^{q}\right)+p\left(\frac{1}{3}-c_{3}\right), \end{array}$$
where $c_{2}=1-c_{1}-c_{3}$ is treated as a dependent variable. The set of Equation (Equation 14) allows us to calculate numerically the temporal evolution of the system and also the stationary states, which are given by the condition:(15)$$\frac{dc_{1}}{dt}=\frac{dc_{3}}{dt}=0.$$

The problem of finding stationary states can also be approached analytically. First, let us notice that an obvious solution is of the form $c_{1}=c_{2}=c_{3}=\frac{1}{3}$. If we insert it in Equation (Equation 14), we see that the stationary condition is satisfied regardless of *p* and *q*. Other solutions can be obtained due to the fact that *p* appears only linearly in the above equations and by noticing the symmetry between states 1 and 3, which is clearly seen from Equation (Equation 14). Although it may be broken by the initial conditions, it is mostly satisfied. Equating the time derivatives (Equation 14) to zero and employing the symmetry $c_{1}=c_{3}$, we obtain a single equation for stationary states other than $c_{1}=c_{2}=c_{3}$. This can of course be expressed in the language of any of the three variables $c_{1},c_{2},c_{3}$. We use the middle opinion because it is somehow dominant, as will be shown later, and thus:(16)$$\begin{array}{c} p=\frac{\left(1-c_{2}^{st}\right){\left(c_{2}^{st}\right)}^{q}-2c_{2}^{st}{\left(\frac{1-c_{2}^{st}}{2}\right)}^{q}}{\left(1-c_{2}^{st}\right){\left(c_{2}^{st}\right)}^{q}-2c_{2}^{st}{\left(\frac{1-c_{2}^{st}}{2}\right)}^{q}-\frac{1}{3}+c_{2}^{st}}, \end{array}$$
where $c_{2}^{st}$ denotes the fraction of opinion 2 in the stationary state. In the next section, we use Equation (Equation 16) to show $c_{2}^{st}$ as a function of *p* simply by flipping the plot, as was done in [44]. Moreover, we use Equation (Equation 14) to obtain phase portraits in plane ($c_{1},c_{3}$).

## 4. Results

### 4.1. Trajectories

Let us start by presenting sample trajectories that can be obtained numerically from Equation (Equation 14). These are valid for arbitrary initial conditions—that is, even if the symmetry $c_{1}=c_{3}$ is not fulfilled. As we see in Figure 1, for a given initial conditions and a fixed size of the influence group *q* different scenarios can be observed, depending on the value of the parameter *p*.

Let us look at the example shown in Figure 1. Here, initially, opinion 1 (let us call it leftists) strongly dominates over opinion 3 (rightists) but only slightly over opinion 2 (centrists). For a small probability of independence, as shown in Figure 1a,b, the system eventually polarizes between leftists and rightists, whereas centrists disappeared almost completely. The temporal evolution in this case is interesting because initially the number of leftists increases, and thus one may think that this option will eventually win. A completely different scenario is observed for larger values of *p* as shown in Figure 1c.

Here, both extreme opinions fade away, and the consensus on the central opinion is reached. Finally, for large values of *p* disagreement $c_{1}=c_{2}=c_{3}$ is reached, see Figure 1d. This single example shows the richness of the behavior that the model exhibits, which was not seen in the model without BC [24]. Without BC, only two final states were possible, independently on *q*: (1) domination of a single opinion, which was initially dominant or (2) disagreement $c_{1}=c_{2}=c_{3}$. In general, for the model with BC, we can distinguish four types of the steady state, which we will refer to later in this paper: **disagreement:**
$c_{1}=c_{2}=c_{3}=1/3$, **central dominance:**$c_{2}>c_{1}=c_{3}$, **extreme dominance:**$c_{1}>c_{3}>c_{2}$ or $c_{3}>c_{1}>c_{2}$ and **polarization:**$c_{1}=c_{3}>c_{2}$.

### 4.2. Stationary States

To analyze the system more systematically, we now focus on stationary states. As long as the symmetry $c_{1}=c_{3}$ is met, all possible stationary points are described by the formula Equation (Equation 16) or $c_{2}=\frac{1}{3}$, as shown in Figure 2a, in which solid lines correspond to stable stationary states, whereas dashed lines correspond to unstable ones. The stability of stationary solutions has been determined, as always, by calculating the eigenvalues of the corresponding Jacobian matrix [52]. The critical point $p^{*}=p^{*}\left(q\right)$, above which the disagreement is always reached, decreases with *q*, similarly to the model without BC [24].

However, the shape of the curve $c_{2}^{st}=c_{2}^{st}\left(p\right)>1/3$ in Figure 2a is very different from the one observed in [24]. Typically, the relationship $p=p(c_{k}^{st})$ takes one of two forms [24,43,44]: (1) has one maximum ($p=p^{*}$), and all solutions along the curve are stable; or (2) has two maxima ($p=p_{up}^{*}$) with a local minimum between them ($p=p_{low}^{*}$), and solutions between maxima are unstable.

In such a typical case, it is easy to determine that the first case corresponds to a continuous phase transition with the critical point at $p=p^{*}$ and the second one to a discontinuous phase transition with an area of metastability (hysteresis) between spinodals $p=p_{low}^{*}$ and $p=p_{up}^{*}$. On the other hand, in Figure 2a we see that $p=p(c_{k}^{st})$, given by Equation (Equation 16), has only one maximum; however, part of the curve represents unstable steady states.

To understand this phenomenon, we must go beyond the symmetry condition $c_{1}=c_{3}$. Therefore, in Figure 2b,c we present the numerical stable stationary solutions obtained from the time evolution of Equation (Equation 12) for three distinct initial conditions $c_{0}=\left(c_{1}\left(0\right),\;c_{2}\left(0\right),\;c_{3}\left(0\right)\right)$, which are shifted from unstable fixed points by a small parameter $\varepsilon =10^{-5}$. Figure 2b shows that, for the initial condition $c_{0}=\;\left(0+\varepsilon ,\;1-\varepsilon ,\;0\right)$, a jump is seen for all q≥2, which corresponds to a discontinuous phase transition, as in the model without BC.

However, the bottom part of Figure 2b looks different from that for the model without BC, for which the results for *q* = 5 are marked with black lines. First, for the model with BC the transition point to the state of disagreement is split into two, depending on the initial conditions. Secondly, as seen better in Figure 2c, which is a zoom in panel (b), there is a second hysteresis for q>5, which appears for small values of *p*.

As for the results shown in Figure 2b, we would also like to draw your attention to the irreversibility of polarization, reported recently in [53]. For the sake of clarification, let us focus on q=2. For $c_{2}\left(0\right)=1$ (initial centrist dominance) with increasing *p* the system goes along the upper branch and undergoes a sharp transition at $p_{up}^{*}∼0.27$, at which $c_{2}$ jumps from 1/2 to 1/3 and then stays at this value for $p>p_{up}^{*}$. However, the reverse path—that is, starting from $c_{2}\left(0\right)=1/3$—is always along the lower branch (polarization), continuously up to $c_{2}=0$, which denotes extreme polarization. Therefore, once the system falls into the lower branch, it can never reach a centrist dominance again and remains polarized. A similar phenomenon called *cusp catastrophe* has been reported for various social systems, both at the level of societies and people [28,54,55].

### 4.3. Phase Portraits

To better understand the behavior presented in Figure 2, we decided to study the phase portraits, obtained from Equation (Equation 14), for different values of the model parameters q,p. For q=1, there is always a disagreement, independently of p>0, analogously to the model without BC [24]. However, already for q=2, while increasing the value of *p*, two bifurcations appear that separate three phases as shown in Figure 3: **(2)**polarization + central dominance, **(1)**disagreement + central dominance and **(0)**disagreement.
We realize that the numbering above from (2) to (0) may be puzzling; however, it is intentional, which will hopefully become clear in the next paragraph.

Qualitatively, the same results are obtained for q=2 and q=3. However, for q=4 extreme dominance is also possible, which was not observed for q=2,3. It appears for small values of *p* and is related to the additional bifurcation that introduces a new phase (3) extreme dominance + central dominance as shown in Figure 4.

For q=5,6,7, the system behavior is even richer, and in total, there are five possible phases separated by the bifurcations as shown in Figure 5. Here, an additional phase appears for which coexistence of three types of steady states is possible, namely: polarization + central dominance + extreme dominance. However, this is not the end of increasing the complexity of system behavior. For q>7, one more phase appears, namely: disagreement + central dominance + extreme dominance.

If we consider the results for all values of *q*, we can distinguish six different phases in total: **(5)**polarization + central dominance + extreme dominance, **(4)**disagreement + central dominance + extreme dominance, **(3)**extreme dominance + central dominance, **(2)**polarization + central dominance, **(1)**disagreement + central dominance and **(0)**disagreement.
Not only the number of phases but also the order of their appearance depends on *q*. As we see in Figure 6, the order of phases with decreasing p>0 is the following:(17)$$\begin{array}{ccc} q=1 & :\; & \left(0\right)\\ q\in \{2,3\} & :\; & \left(0)\to (1)\to (2\right)\\ q=4 & :\; & \left(0)\to (1)\to (2)\to (3\right)\\ q\in \{5,6,7\} & :\; & \left(0)\to (1)\to (2)\to (5)\to (3\right)\\ q\in \{8,9,10\} & :\; & \left(0)\to (1)\to (4)\to (5)\to (3\right) \end{array}$$
Only in phase (0) there is no coexistence of different types of steady states. In all the others, several of them can be achieved, depending on the initial conditions. This means that social hysteresis exists for all values of q>1 and $p\in (0,p_{up}^{*})$, where $p_{up}^{*}=p_{up}^{*}\left(q\right)$ is the bifurcation above which only phase (0) exists. As seen in Figure 3, Figure 4 and Figure 5, this is a saddle-node bifurcation during which stable and unstable fixed points collide and annihilate. As always in the noisy *q*-voter models $p_{up}^{*}=p_{up}^{*}\left(q\right)$ is a decreasing function of *q* [24,43,44], which is clearly seen in Figure 6.

## 5. Discussion

It was recently shown that whether opinions are discrete or continuous depends on the agent’s attention to the given issue: binary in the case of high attention or continuous in other cases [55]. However, three-state opinions, as considered in this paper, can be useful when describing an actual choice between two extremes and balanced option or the response on the three-point psychometric Likert scale (disagree, neither agree nor disagree and agree).

This scale was recently tested in a simple discrete choice experiment, in which several granularities of the response scales (2, 3, 4 and 5 points) were compared [56]. As a result, it was concluded that odd-numbered Likert scales contribute to lower error variance, and using 3-point response scales seems more advisable than using 5-point ones. Therefore, three-state variables seem to be particularly useful when it comes to measuring opinions in social surveys.

This motivated us to focus on the three-state model with BC. This is not the first paper to be devoted to this issue. However, most of the earlier models, often based on a linear voter model [9,10,14,19,27] or on the majority vote model [25,26,29,30,31,57], did not allow us to study directly the effect of influence group size on the emergence on different social collective phenomena.

It is likely that the only other sociophysics model—apart from the *q*-voter model—that allows one to directly examine the role of the size of the group is the Galam model, which has also been studied in its three-state version [12,16]. However, the Galam model, which is a hierarchical voting model, is not devoted to analyze various collective phenomena that appear spontaneously under the influence of interactions in continuous time (sequential updates). It is rather to design a winning strategy as an outcome of the voting, which occurs in discrete time steps (synchronous updates).

Of course, modifications of the update scheme to random sequential updating is possible and has already been proposed in the case of binary opinions [58]. However, to the best of our knowledge, the Galam model has never been analyzed under the constraint of BC. In the future, one could generalize the sequential version of the model proposed in [58] to multi-state opinions and then introduce BC, which seems to be an interesting task.

Thus far, there are two three-state models in the context of which our model should be discussed. The first is based on a linear voter model [9,10]. Within this model, in a single update, a randomly selected agent adopts the opinion of a randomly selected neighbor if the opinions of the agent and the neighbor do not differ by more than one. In such a model, the influence group is always of size 1, and thus the role of the influence group cannot be studied.

Moreover, due to the lack of noise, a system always evolves toward an absorbing (frozen) state. Depending on the initial fraction of centrists, one of two absorbing states can be reached: consensus of any of the three opinions or polarization. Our model reduces to this for q=1 and p=0, and thus it can be treated as a generalization of the model introduced in [9]. However, we find the results for q>1 and p>0 particularly interesting. For these parameter values, our model displays social hysteresis, which has recently received particular attention in the context of social polarization [53].

The second three-state model with BC, which is crucial in the context of this paper, is based, like ours, on a multi-state noisy qVM [36]. The only difference from our model is that, additionally, the restriction on changing opinions under the influence of independence was introduced. In our model, BC is only about the interaction between agents, i.e., it applies to conformity. In the case of independence, any of three opinions can be taken, which corresponds exactly to the noise in the original multi-state *q*-voter model [24]. On the contrary, in [36], also under independence, the agent could not change her opinion by more than one.

This seemingly small difference between the models has surprisingly far-reaching effects. In particular, in our model, disagreement is possible, as in [24]. On the other hand, the complete disagreement, defined as $c_{1}=c_{2}=c_{3}$, is never possible in the model proposed in [36]. Even for p=1, when the system is driven only by the noise, centrists slightly dominate—that is, $c_{1}=c_{3}=2/7,\;c_{2}=3/7$. This is because, as written in [36]: *transitions between states 1 and 3 are forbidden; therefore, opinions 1 and 3 have only opinion 2 as a neighboring one, whereas for opinion 2, both 1 and 3 are the neighboring states*.

The lack of complete disagreement is certainly not the main difference with our model. One can always manipulate the transition rates in the case of independence in such a way that complete disagreement becomes possible. However, more interestingly, in [36], polarization between extreme opinions was not observed. In general, the model studied here shows a much richer behavior than the one introduced in [36]. This shows how a small change, introduced at the microscopic level, can dramatically alter and enrich the macroscopic behavior of a model. However, a heuristic understanding of what precisely influences the polarization and when irreversible polarization can be seen in this type of models is still missing and is a desirable task for the future.

## Figures and Tables

**Figure 1 entropy-24-00983-f001:**
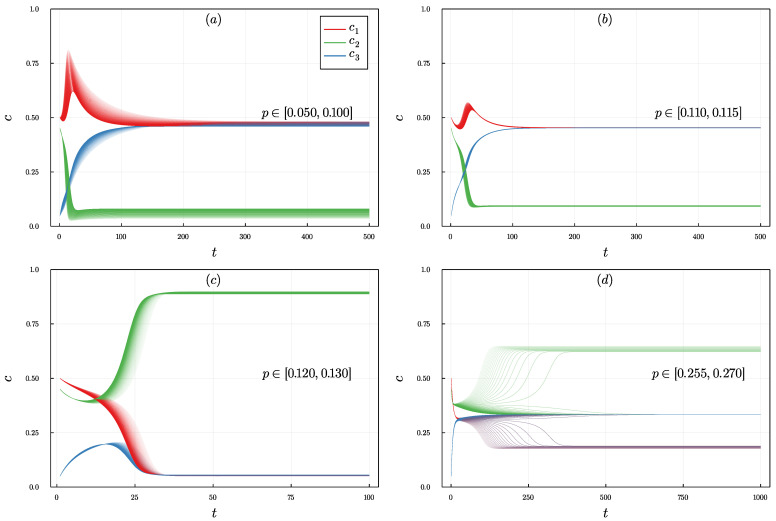
Trajectories showing the evolution of opinion concentration $c\in \{c_{1},c_{2},c_{3}\}$, as indicated by the legend, over time *t* for the size of the influence group q=2 from the initial condition $c_{1}\left(0\right)=0.5,$$c_{2}\left(0\right)=0.45,$$c_{3}\left(0\right)=0.05$. Each panel shows trajectories for values of the probability of independence *p* from different ranges: (**a**) p∈[0.050,0.100], (**b**) p∈[0.110,0.115], (**c**) p∈[0.120,0.130], (**d**) p∈[0.255,0.270]. The color intensity of the trajectories in each panel increases with *p*.

**Figure 2 entropy-24-00983-f002:**
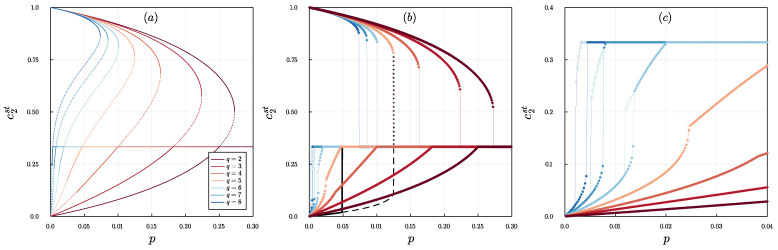
Stationary concentration of agents in the middle state as a function of the probability of independence *p* for several values of *q*, as indicated in the legend on the left panel: (**a**) Analytical solutions obtained within Equation (Equation 16) for the condition $c_{1}=c_{3}$—solid and dashed lines correspond to stable and unstable steady states, respectively. (**b**) Numerical stable solutions obtained from the time evolution of Equation (Equation 14) for three different initial conditions $c_{0}=\left(c_{1}\left(0\right),\;c_{2}\left(0\right),\;c_{3}\left(0\right)\right)$: $c_{0}=\left(\frac{1}{2}-\varepsilon ,\;0+\varepsilon ,\;\frac{1}{2}\right)$ marked by empty circles, $c_{0}=\;\left(1-\varepsilon ,\;0+\varepsilon ,\;0\right)$ marked by full circles and $c_{0}=\;\left(0+\varepsilon ,\;1-\varepsilon ,\;0\right)$ marked by diamonds. (**c**) Zoom of the panel. (**b**) The initial points $c_{0}$ were moved from unstable fixed points by a small parameter $\varepsilon =10^{-5}$. In panel (**b**), additional black lines indicate the results of the model without BC, as studied in [24], for q=5 and the same three types of initial conditions as for the model with BC: $c_{0}=\left(\frac{1}{2}-\varepsilon ,\;0+\varepsilon ,\;\frac{1}{2}\right)$—solid line, $c_{0}=\;\left(1-\varepsilon ,\;0+\varepsilon ,\;0\right)$—dashed line, and $c_{0}=\;\left(0+\varepsilon ,\;1-\varepsilon ,\;0\right)$—dotted line.

**Figure 3 entropy-24-00983-f003:**
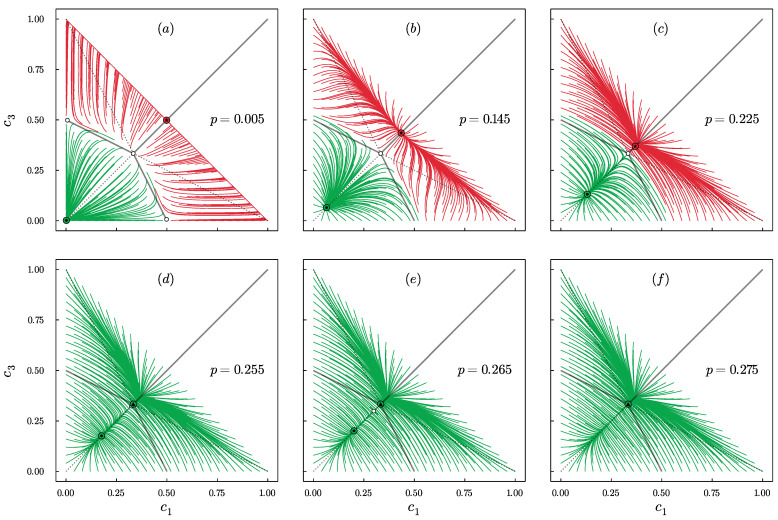
Phase portraits for q=2 and several values of *p*, as indicated in the panels: *p* increases from panel (**a**) to (**f**). Stable points are denoted by the circles with small markers inside: (▲) disagreement, (•) central dominance and (★) polarization. Unstable fixed points are denoted by the small empty circles. Trajectories associated with different types of attraction basins are highlighted with different colors. For q=2, three different phases are visible: (**a**–**c**) polarization + central dominance, (**d**,**e**) disagreement + central dominance and (**f**) disagreement.

**Figure 4 entropy-24-00983-f004:**
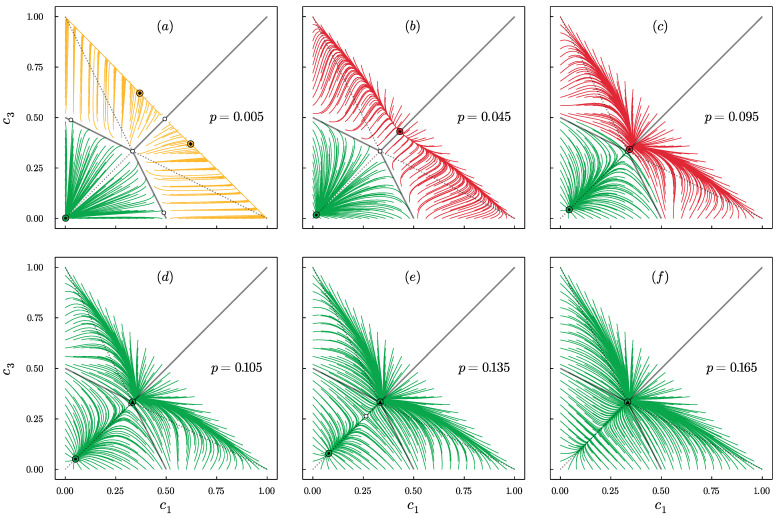
Phase portraits for q=4 and several values of *p*, as indicated in the panels: *p* increases from panel (**a**) to (**f**). Stable points are denoted by the circles with small markers inside: (▲) disagreement, (•) central dominance, (◆) extreme dominance and (★) polarization. Unstable fixed points are denoted by the small empty circles. Trajectories associated with different types of attraction basins are highlighted with different colors. For q=4 four different phases are visible: (**a**) extreme dominance + central dominance, (**b**,**c**) polarization + central dominance, (**d**,**e**) disagreement + central dominance and (**f**) disagreement.

**Figure 5 entropy-24-00983-f005:**
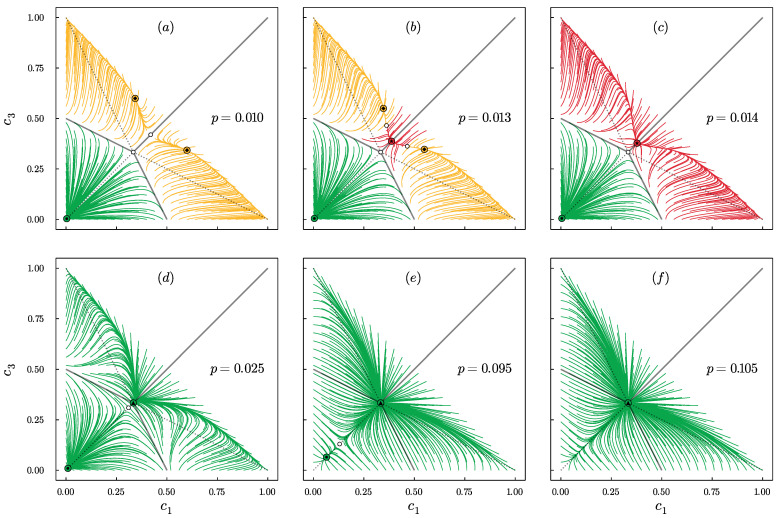
Phase portraits for q=6 and several values of *p*, as indicated in the panels: *p* increases from panel (**a**) to (**f**). Stable points are denoted by the circles with small markers inside: (▲) disagreement, (•) central dominance, (◆) extreme dominance and (★) polarization. Unstable fixed points are denoted by the small empty circles. Trajectories associated with different types of attraction basins are highlighted with different colors. For q=6 five different phases are visible: (**a**) extreme dominance + central dominance, (**b**) polarization + central dominance + extreme dominance, (**c**) polarization + central dominance, (**d**,**e**) disagreement + central dominance and (**f**) disagreement.

**Figure 6 entropy-24-00983-f006:**
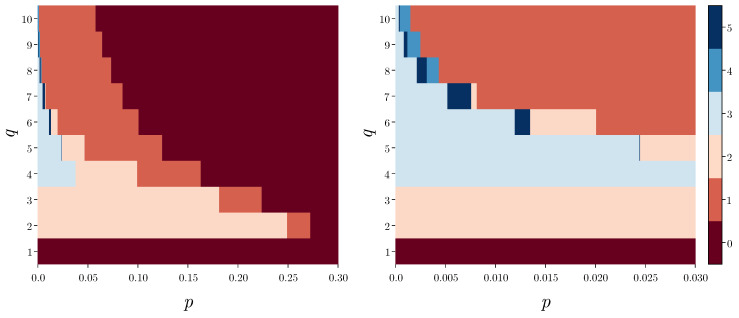
Phase diagrams showing the different phases that occur for given values of model parameters q∈{1,10} and p∈(0,0.3) (left panel), zoom for p∈(0,0.03) (right panel). Numbers at the color bars indicate the following phases: (0) disagreement, (1) disagreement + central dominance, (2) polarization + central dominance, (3) extreme dominance + central dominance, (4) disagreement + central dominance + extreme dominance and (5) polarization + central dominance + extreme dominance.

## Data Availability

Not applicable.

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
