# Peer review of "Consensus, Polarization and Hysteresis in the Three-State Noisy q-Voter Model with Bounded Confidence"

_entropy, 2022, doi:10.3390/e24070983_

Round 1

Reviewer 1 Report

In the manuscript, the authors study a 3-state noisy q-voter model with bounded confidence through analytical and numerical calculations. The results and discussion are interested. Before the manuscript can be recommended for publication, the authors need to consider some comments:

# In the abstract, the authors say “domination of centrists (consensus)”. I did not understand such definition of consensus, since usually the consensus is defined by all agents in the network sharing the same opinion. The definition is important since in page 6 the authors discuss about Figure 1(c). In such a figure, there are small but nonzero fractions of opinions 1 and 3, and a dominance of centriststs, i.e., no all agents in the population are centrists. Indeed, after that, in line 166 (page 7), the authors call that situation as “central dominance”, not “consensus”.

# Line 191: a jump is seen for all q > 2”, is it correct? Or the correct should be q >= 2 ?

# In the beginning of page 12, the authors say: “However, the Galam model, which is a hierarchical voting model, is not devoted to analyze various collective phenomena that appear spontaneously under the influence of interactions in continuous time (sequential updates).“

However, collective phenomena in the Galam models under the influence of distinct interactions, namely inflexibility and independence, was analyzed in the paper:

Inflexibility and independence: Phase transitions in the majority-rule model, Physical Review E 92, 062122 (2015)

In this paper, in addition to the model considered in the complete graph, the authors also analyzed the Galam model with independence and inflexibility in lattices, and random sequential updates were considered. This paper should me mentioned in the conclusion of the manuscript.

Another comment: after that, the authors say “Moreover, to the best of our knowledge, the Galam model has never been analyzed under the constraint of BC.” Indeed, I also never saw the galam model analyzed considering BC. It would be an interesting work for the future.

# Typo:

line 45: characteristic characteristic feature

Author Response

Reviewer #1: We are grateful for the thorough report, which contributed to the improvement of our paper. The complete list of all changes and our replies to your comments are provided below. All changes in the manuscript have been highlighted in blue for your convenience.

  1. In the abstract, the authors say “domination of centrists (consensus)”. I did not understand such definition of consensus, since usually the consensus is defined by all agents in the network sharing the same opinion. The definition is important since in page 6 the authors discuss about Figure 1(c). In such a figure, there are small but nonzero fractions of opinions 1 and 3, and a dominance of centriststs, i.e., no all agents in the population are centrists. Indeed, after that, in line 166 (page 7), the authors call that situation as “central dominance”, not “consensus”.

Indeed in many models of opinion dynamics, consensus is understood as a unanimity, but mostly in the models without the noise. However, from the social point of view, a consensus refers to group decision-making processes on establishing agreement of the supermajority, which differentiates consensus from unanimity. In the revised version we have commented on this in the abstract and introduction. Moreover, we clarified that we use word consensus and domination  interchangeably.

  1. # Line 191: “a jump is seen for all q > 2”, is it correct? Or the correct should be q >= 2 ?

Thank you for your perceptiveness - corrected

  1. # In the beginning of page 12, the authors say: “However, the Galam model, which is a hierarchical voting model, is not devoted to analyze various collective phenomena that appear spontaneously under the influence of interactions in continuous time (sequential updates).“ However, collective phenomena in the Galam models under the influence of distinct interactions, namely inflexibility and independence, was analyzed in the paper: Inflexibility and independence: Phase transitions in the majority-rule model, Physical Review E 92, 062122 (2015). In this paper, in addition to the model considered in the complete graph, the authors also analyzed the Galam model with independence and inflexibility in lattices, and random sequential updates were considered. This paper should me mentioned in the conclusion of the manuscript.

Thank you for reminding us of this work, which of course we are familiar with and have cited in the past, but somehow overlooked here. We have added a citation and a comment in the discussion.

  1. Another comment: after that, the authors say “Moreover, to the best of our knowledge, the Galam model has never been analyzed under the constraint of BC.” Indeed, I also never saw the galam model analyzed considering BC. It would be an interesting work for the future.

We are glad that the reviewer is also not familiar with such work. We also believe that it would be interesting work for the future, especially in the context of the paper mentioned above.

  1. # Typo: line 45: characteristic characteristic feature

Thank you for your perceptiveness - corrected

Reviewer 2 Report

Overall the paper is a technically sound and interesting piece of work. Opinion dynamics is a topic of ever growing interest, given the potential of applications in a broad range of fields. The authors introduce a new model (which is a minor modification of existing ones) and analyse it in detail. However, I believe the paper has some major issues as it currently stands which the authors should address.

First, I personally found the introduction very confusing. Specifically, the authors start by providing a report of how many papers have been published on this topic, and I fail to see why this is relevant. The fact that many people are publishing in the same field is not good enough a motivation to come up with a new model. In fact, it might be the other way around. Also, the authors only briefly mention the research their work is built upon. For instance, they mention their model is an extension of the qVM but do not explain what the qVM is and why this is important. I only found out in the Model section what they exactly intended to do and how this was similar/different to existing models.

Second, precisely because there are so many new models of opinion dynamics being published every year, the authors need to justify why yet another one is needed. Linking it to concrete examples and applications, for instance, where existing models would fail to capture the dynamics of the system or its full complexity.

Third, the discussion section is more suitable to be the introduction! The authors describe other models and compare them with theirs, but do not seem to discuss their results and the consequent implications at all.

Finally, the only concern I have about the methodology: the authors use a fully connected network to perform their analysis. Although this is customary in opinion dynamics, it also make this model unrealistic for most purposes. For this reason, I would either like to see some small experiments with realistic network topologies or a very good explanation as to why only using a fully connected network is ok.

To conclude, I believe the model is interesting and definitely has merit. However, this does not come out in any way in the paper. I believe the authors should considerably rewrite the introduction and discussion sections to improve their arguments and the overall clarity of the paper, as well as address the remaining points above, before this can be considered for publication.

Author Response

Reviewer #2: We are grateful for the thorough report, which contributed to the improvement of our paper. The complete list of all changes and our replies to your comments are provided below. All changes in the manuscript have been highlighted in blue for your convenience.

  1. First, I personally found the introduction very confusing. Specifically, the authors start by providing a report of how many papers have been published on this topic, and I fail to see why this is relevant. The fact that many people are publishing in the same field is not good enough a motivation to come up with a new model. In fact, it might be the other way around. Also, the authors only briefly mention the research their work is built upon. For instance, they mention their model is an extension of the qVM but do not explain what the qVM is and why this is important. I only found out in the Model section what they exactly intended to do and how this was similar/different to existing models.

Thank you for this comment, as it made us realize that we have not pointed out clearly enough the reason why we are introducing the model. Therefore we have added two paragraphs in the introduction.

  1. Second, precisely because there are so many new models of opinion dynamics being published every year, the authors need to justify why yet another one is needed. Linking it to concrete examples and applications, for instance, where existing models would fail to capture the dynamics of the system or its full complexity.

The same as above – we have modified the introduction and we hope now it is clear.

  1. Third, the discussion section is more suitable to be the introduction! The authors describe other models and compare them with theirs, but do not seem to discuss their results and the consequent implications at all.

We have slightly reformulated summary and together with the modified introduction, we hope that now it is satisfactory.

  1. Finally, the only concern I have about the methodology: the authors use a fully connected network to perform their analysis. Although this is customary in opinion dynamics, it also make this model unrealistic for most purposes. For this reason, I would either like to see some small experiments with realistic network topologies or a very good explanation as to why only using a fully connected network is ok.

To be honest, in this case we thought we gave a very good justification for why we were using the full graph structure, and it is given at the beginning of section 3. The idea was mainly to study the influence of BC, and the multistate q-voter model has so far been studied only on the full graph. However, a comment on this issue has been added in the new version, right after original justification in section 3.

Reviewer 3 Report

In this article the authors study the three-state noisy q-voter model with bounded interactions, with the aim of exploring how the size of the influence group affects the emergence of collective phenomena such as consensus, polarization and extremism.  With probability p an agent takes one of the three opinions at random (independence), and with probability 1-p adopts the opinion of an unanimous group of q random neighbors (conformity) only if the opinion difference between the agent and the group is less than 2, that is, interactions between leftists and rightists are forbidden.  The introduction of this bounded confidence (BC) is the special feature of the model, as compared to the multi-state noisy VM studied in ref. [24].  The model is also very similar to the one studied in ref. [36], but this work also applies BC to the independence mechanism.  The authors show that applying BC in both, conformity and independence, results in a very rich new phenomenology not seen in the related studies mentioned above [24, 36], as for example the appearance of new phases.

The article is well written, correctly organized and scientifically sound.  The model is analyzed in great detail within a mean-field approach, showing all possible behaviors for the parameters p and q.  Calculations are all correct as far as I could check.  For these reasons, I recommend the article for publication.  However, there are a few points that the authors should consider to improve the article.

- The definition of the model is not clear. In line 103, it says that m_q=1,2,3 means that the q-panel is unanimous, where m_q is the mean opinion of the q-panel ["mean" is missing in point b)].  However, if for instance q=3 and the opinions of the three agents are 1, 2 and 3, then m_q=2, but there is no unanimity.  I understand that the unanimity condition is simply that the opinions of the q agents are the same (S_j=S for all j=1,..,q), so I do not see why they need to calculate m_q.  Therefore, I believe that the condition for a conformity interaction should be revised.

- Also, the BC condition could be simply written as agents with opinions 1 and 3 do not interact.

- Conclusions: it is mentioned that in the model of ref. [36] disagreement is not possible, and centrists always dominate.  I do not understand why because, although with BC, the model in [36] also has independence (noise), and so I would expect disagreement for p large enough.  It also mentioned that polarization is not observed in [36], and I do not see why.  I understand that in the model of [36] opinion changes during an independence event are between states 1 and 2, and between states 2 and 3 (changes from 1 and 3, and vice-versa are not allowed).  What is special about this slight change with respect to the present model, where also noise transitions 1<-->3 are possible, which results in the absence of polarization and disagreement?  The authors should add a sentence providing an insight into this.

- Figs. 3, 4 and 5: to read the plots more easily, it might help to add labels next to the stable fixed points: For instance C (central dominance), P (polarization), E (extreme dominance), D (disagreement).  This might help identify the 6 different phases described in page 9.  For instance, I understand that Fig. 5(b) has the points P, C and E, describing the phase 5: polarization + central dominance + extreme dominance.

-  From line 178, it is explained that the shape of c2(p) in ref. [24] is very different from the one fund in the model.  Given that describing this shape is quite technical, perhaps they should add an schematic plot of c2(p) in [24].

- I found the discontinuous transitions shown in Fig. 2 very interesting.  For q=2, starting from c2=1 (centrist dominance) for p=0 and increasing p, the system goes along the upper branch and undergoes a sharp transition at p*~0.27 where c2 jumps from 0.5 to the value 0.33 in the lower branch, and then c2 keeps along the stable line c2=0.33.  However, the reverse path from p=1 to p=0 is always along the lower branch until c2=0 (polarization).  Therefore, once the system falls into the lower branch it can never reach a centrist dominance again.  A similar phenomenon called "cusp catastrophe" is observed in some disciplines as for instance ecology (the system is never recovered once a threshold is overcome). Perhaps the authors might want to check in the theory of dynamical systems if the cusp catastrophe has a relation with the transition found in the model.

- Just before Eq. (2), it should be "concentration of opinion k agents" instead of "social opinions".

- Line 158: "consensus on the central opinion is reached".  In Fig. 1(c) the consensus is not full (c2 ~ 0.9), so perhaps it should say a quasi-consensus or an almost full consensus.

- Caption of Fig. 2: initial conditions are non-physical because they do not obey the conservation relation c1+c2+c3=1.  Symbols in panels (b) and (c) should be larger because it is hard to distinguish between empty circles, full circles and diamonds. More space between symbols could help.

- Line 192: should it be "q >= 2" instead of "q>2"?

- Caption of Fig. 6: phase 3 should be "extreme dominance + central dominance".

Author Response

Reviewer #3: We are grateful for the thorough report, which contributed to the improvement of our paper. The complete list of all changes and our replies to your comments are provided below. All changes in the manuscript have been highlighted in blue for your convenience.

  1. The definition of the model is not clear. In line 103, it says that m_q=1,2,3 means that the q-panel is unanimous, where m_q is the mean opinion of the q-panel ["mean" is missing in point b)].  However, if for instance q=3 and the opinions of the three agents are 1, 2 and 3, then m_q=2, but there is no unanimity.  I understand that the unanimity condition is simply that the opinions of the q agents are the same (S_j=S for all j=1,..,q), so I do not see why they need to calculate m_q.  Therefore, I believe that the condition for a conformity interaction should be revised.

Thank you for your perceptiveness. You are completely right. We got rid of m_q and are simply writing about unanimity.

  1. Also, the BC condition could be simply written as agents with opinions 1 and 3 do not interact.

You are right that this is simple and elegant definition of BC and probably clear for many readers. However, we were afraid that it would be not clear enough in the case of the q-voter model for people that are not familiar with the model. Moreover, the advantage of our more detailed description is that it can be easily implemented in the multistate models. Therefore, in the revised version of the manuscript we kept the old description of BC, but we also added: “,which  means that agents with opinions 1 and 3 do not interact”

  1. Conclusions: it is mentioned that in the model of ref. [36] disagreement is not possible, and centrists always dominate.  I do not understand why because, although with BC, the model in [36] also has independence (noise), and so I would expect disagreement for p large enough.  It also mentioned that polarization is not observed in [36], and I do not see why.  I understand that in the model of [36] opinion changes during an independence event are between states 1 and 2, and between states 2 and 3 (changes from 1 and 3, and vice-versa are not allowed).  What is special about this slight change with respect to the present model, where also noise transitions 1<-->3 are possible, which results in the absence of polarization and disagreement?  The authors should add a sentence providing an insight into this.

You are right that stating simply that disagreement was impossible in Radosz & Doniec (2021) was unclear and needed clarification. In the revised version we write precisely: On the other hand, the complete disagreement, defined as $c_1=c_2=c_3$, is  never possible in the model proposed in Radosz & Doniec (2021). Even for $p=1$, when the system is driven only by the noise, centrists slightly dominate, that is $c_{1}=c_{3}=2/7,  c_{2}=3/7$. This is because, as written in Radosz & Doniec (2021): transitions between states 1 and 3 are forbidden, so opinions 1 and 3 have only opinion 2 as a neighbouring one, whereas for opinion 2 both 1 and 3 are the neighbouring states.} However, heuristic understanding what precisely influence the polarization and when irreversible polarization can be seen in this type of models is still missing and is desirable task for the future.

  1. Figs. 3, 4 and 5: to read the plots more easily, it might help to add labels next to the stable fixed points: For instance C (central dominance), P (polarization), E (extreme dominance), D (disagreement).  This might help identify the 6 different phases described in page 9.  For instance, I understand that Fig. 5(b) has the points P, C and E, describing the phase 5: polarization + central dominance + extreme dominance.

Referee is right that denoting these points with letters would be helpful and we tried to do it. However, this looked neither aesthetically pleasing nor transparent, due to the large number of trajectories visible in the figure, which we did not want to remove. So instead, we added small markers, different for each type of social structure, in the middle of the circles denoting the fixed points. We realize that markers may not be clear enough, so we have also added descriptions of what phases are visible on each panel (a)-(f).

  1. From line 178, it is explained that the shape of c2(p) in ref. [24] is very different from the one fund in the model.  Given that describing this shape is quite technical, perhaps they should add an schematic plot of c2(p) in [24].

We have added such a plot in Fig 2(b).

  1. I found the discontinuous transitions shown in Fig. 2 very interesting.  For q=2, starting from c2=1 (centrist dominance) for p=0 and increasing p, the system goes along the upper branch and undergoes a sharp transition at p*~0.27 where c2 jumps from 0.5 to the value 0.33 in the lower branch, and then c2 keeps along the stable line c2=0.33.  However, the reverse path from p=1 to p=0 is always along the lower branch until c2=0 (polarization).  Therefore, once the system falls into the lower branch it can never reach a centrist dominance again.  A similar phenomenon called "cusp catastrophe" is observed in some disciplines as for instance ecology (the system is never recovered once a threshold is overcome). Perhaps the authors might want to check in the theory of dynamical systems if the cusp catastrophe has a relation with the transition found in the model.

We are very grateful for this comment. We included it in the revised version of the paper and cited several papers, in which cusp catastrophe has been reported for various social systems, both at the level of societies and people \cite{Sob:12,Sob:16,Maa:Dal:Wal:20}. The last reference was present already in the original version of the paper, and first two was added in the revised version of the manuscript.

  1. Just before Eq. (2), it should be "concentration of opinion k agents" instead of "social opinions".

You are right – we have corrected this.

  1. Line 158: "consensus on the central opinion is reached".  In Fig. 1(c) the consensus is not full (c2 ~ 0.9), so perhaps it should say a quasi-consensus or an almost full consensus.

The issue of consensus was also noted by another reviewer, so in the new version of the paper we define consensus precisely in the Introduction: It is worth noting here that in this work

we use the term consensus as defined by Webster’s or Oxford dictionary, which define it

as a general agreement and distinguish it from unanimity. In fact, consensus, understood

as unanimity, is almost impossible to achieve in large social group. It only appears in

theoretical models in the absence of noise. In the presence of noise, only strong dominance

of one opinion is possible [29, 43 ], which basically agrees with the actual definition of

consensus. Therefore, in the rest of the paper we use the words consensus and dominance

of one opinion interchangeably.

  1.  Caption of Fig. 2: initial conditions are non-physical because they do not obey the conservation relation c1+c2+c3=1.  Symbols in panels (b) and (c) should be larger because it is hard to distinguish between empty circles, full circles and diamonds. More space between symbols could help.

Thank you for your perceptiveness. We have corrected the description of the starting points and increased the size of the symbols slightly. We could not make them too large, although we also prefer large readable symbols, because in this case it was important to preserve the density of points to show hysteresis.

Reviewer 4 Report

This manuscript studies an opinion model with three states (left, center and right) and different mechanisms driving opinion changing (independence + q-voter + bounded confidence). The manuscript is well-written, and the results are interesting for the community working on opinion dynamics. Furthermore, I think that the figures and visualization of the results, together with the general presentation of the different phases/states, is particularly appealing. For these reasons, I recommend publishing the manuscript as it is. 

Author Response

Thank you for the positive recommendation. We are happy that you like our paper.

Round 2

Reviewer 1 Report

The authors addressed the points I mentioned in the previous report. The paper can be recommended for publication.

Author Response

Thank you for your positive recommendation.

Reviewer 2 Report

I am happy with the changes the authors made.

Author Response

(The authors gave the same response as above.)

Reviewer 3 Report

The authors have considered and answered satisfactorily most of my concerns.  However, prior acceptance of the article there are some minor issues that they should consider.

1) Definition of the model: although one can guess that unanimity means that all q agents share the same opinion, it should be clearly stated mathematically that the q-panel is unanimous if S_j=S for all j=1,..,q.

3) Conclusions: I am still not convinced that complete disagreement, defined as c_1=c_2=c_3, is never possible in the model proposed in Radosz & Doniec (2021), given that the stationary state depends on how the transition rates of independence are defined.  When p=1, the stationary concentrations c_1=c_3=2/7, c_2=3/7 are obtained when the transitions from 1 to 2 and from 1 to 1 are equal to ½, that is, w_12=½=w_11, and similarly for transitions from 3 to 2, and from 3 to 3, w_32=w_33=½.  The rest of the transitions would be w_21=w_22=w_23=â…“.  As these transition probabilities are arbitrarily, one can also use w_12=â…“, w_11=â…”, w_32=â…“, w_33=â…”, and w_21=w_22=w_23=â…“.  In this case the stationary concentrations correspond to those of complete disagreement c_1=c_2=c_3=1/3.

The following points of the previous report were not answered:

- Line 192: should it be "q >= 2" instead of "q>2"?

- Caption of Fig. 6: phase 3 should be "extreme dominance + central dominance".

Author Response

Thank you very much for all comments and suggestions and please find our reply to the resport below. We have highlighted in blue all changes made in the manuscript.

1) Definition of the model: although one can guess that unanimity means that all q agents share the same opinion, it should be clearly stated mathematically that the q-panel is unanimous if S_j=S for all j=1,..,q.

Thank you for this suggestion. We have corrected the description according to your suggestions.

3) Conclusions: I am still not convinced that complete disagreement, defined as c_1=c_2=c_3, is never possible in the model proposed in Radosz & Doniec (2021), given that the stationary state depends on how the transition rates of independence are defined. 

You are right that one can always manipulate the transition rates but in the original formulation of the model they were defined just as described in our manuscript, which is also consistent with the idea of noise, as introduced usually in the noisy voter models. If we change these probabilities, as you suggest (w_12=â…“, w_11=â…”, w_32=â…“, w_33=â…”) it would mean that keeping the old state is more likely than choosing any other. Of course such a change would influence not only results for p=1 but for all other values of p. However, such a generalized model could be also introduced, but it was not so far.  We commented shortly on this issue writing:

The lack of the complete disagreement is certainly not the main difference with our model. One can always manipulate the transition rates in the case of independence in such a way that the complete disagreement becomes possible. However, more interestingly, in \cite{Rad:Don:21} polarization between extreme opinions is not observed.

Overlooked previously

1) Line 192: should it be "q >= 2" instead of "q>2"?

Yes, indeed it should be "q >= 2" and we corrected this already in the previous version of the manuscript. However due to the changes in other parts of the paper, the line corresponing to this sentence changed from 192 to 209. Once again we thank the Reviewer for this thorough observation.

2)  Caption of Fig. 6: phase 3 should be "extreme dominance + central dominance".

Corrected